# Presence of Multiple Genetic Mutations Related to Insecticide Resistance in Chinese Field Samples of Two Phthorimaea Pest Species

**DOI:** 10.3390/insects15030194

**Published:** 2024-03-14

**Authors:** Jiang Zhu, Ruipeng Chen, Juan Liu, Weichao Lin, Jiaxin Liang, Ralf Nauen, Suhua Li, Yulin Gao

**Affiliations:** 1Shenzhen Branch, Guangdong Laboratory of Lingnan Modern Agriculture, Key Laboratory of Synthetic Biology, Ministry of Agriculture and Rural Affairs, Agricultural Genomics Institute at Shenzhen, Chinese Academy of Agricultural Sciences, Shenzhen 518000, China; zhujiang@caas.cn (J.Z.); lj961527860@163.com (J.L.); lenolin429@163.com (W.L.); 18016192917@163.com (J.L.); 2State Key Laboratory for Biology of Plant Diseases and Insect Pests, Institute of Plant Protection, Chinese Academy of Agricultural Sciences, Beijing 100193, China; m18238756637@163.com; 3Crop Science Division, R&D, Pest Control, Bayer AG, Alfred-Nobel-Str. 50, 40789 Monheim, Germany; ralf.nauen@bayer.com

**Keywords:** *Phthorimaea operculella*, *Phthorimaea absoluta*, target resistance, Ace1, VGSC, RDL

## Abstract

**Simple Summary:**

Insecticide resistance stands as one of the most notable evolutionary phenomena for researchers. Two closely related species pests, *Phthorimaea operculella* and *Phthorimaea absoluta*, both feeding on potato crops, have developed distinct resistance mechanisms. In this study, we compared the presence of target-site mutations in *P. operculella* and *P. absoluta* in three common insecticide targets, Ace1 (acetylcholinesterase), VGSC (voltage-gated sodium channel), and RDL (GABA-gated chloride channel). Unexpected genetic divergence in target-site mutations was observed between the two species. *P. absoluta* had three Ace1 mutations (A201S, L231V, and F290V), four VGSC mutations (M918T, L925M, T928I, and L1014F), and one RDL mutation (A301S). On the other hand, *P. operculella* had Ace1 mutations (F158Y, A201S, and L231V) and only one VGSC mutation (L1014F) at lower frequencies, with no mutations detected in RDL. These findings deepen our understanding of evolutionary dynamics in pest species and offer potential strategies for more precise and sustainable pest control in potatoes.

**Abstract:**

Potatoes hold the distinction of being the largest non-cereal food crop globally. The application of insecticides has been the most common technology for pest control. The repeated use of synthetic insecticides of the same chemical class and frequent applications have resulted in the emergence of insecticide resistance. Two closely related pests that feed on potato crops are the potato tuber moth, *Phthorimaea operculella*, and the tomato leafminer, *Phthorimaea absoluta* (syn. *Tuta absoluta*). Previous studies indicated the existence of insecticide resistance to various classes of insecticides including organophosphates, carbamates, and pyrethroids in field populations of *P. operculella* and *P. absoluta*. However, the exact mechanisms of insecticide resistance in *P. operculella* and to a lesser extent *P. absoluta* remain still poorly understood. Detecting resistance genotypes is crucial for the prediction and management of insecticide resistance. In this study, we identified multiple genetic mutations related to insecticide resistance in two species of *Phthorimaea*. An unexpected genetic divergence on target-site mutations was observed between *P. operculella* and *P. absoluta*. Three mutations (A201S, L231V, and F290V) in Ace1 (acetylcholinesterase), four mutations (M918T, L925M, T928I, and L1014F) in VGSC (voltage-gated sodium channel), and one mutation (A301S) in RDL (GABA-gated chloride channel) have been detected with varying frequencies in Chinese *P. absoluta* field populations. In contrast, *P. operculella* field populations showed three mutations (F158Y, A201S, and L231V) in Ace1, one mutation (L1014F) in VGSC at a lower frequency, and no mutation in RDL. These findings suggest that pyrethroids, organophosphates, and carbamates are likely to be ineffective in controlling *P. absoluta*, but not *P. operculella*. These findings contributed to a deeper understanding of the presence of target-site mutations conferring resistance to commonly used (and cheap) classes of insecticides in two closely related potato pests. It is recommended to consider the resistance status of both pests for the implementation of resistance management strategies in potatoes.

## 1. Introduction

Potatoes (*Solanum tuberosum* L.) rank as the fourth most important food crop globally, next to rice, wheat, and maize [1]. It is a critical crop and widely cultivated in terms of its rich nutritive value. China, the biggest producer of potatoes since 1993 [2], is playing an increasing role in the global potato market. FAO data show that it continues to increase in its importance with an annual production of potatoes in China up to approximately 10 million tons in 2022 (http://www.fao.org/statistics/en/, accessed on 13 March 2023). Most of the potato production takes place in the northern and southwest regions of China, notably in Yunnan province.

The potato tuber moth, *Phthorimaea operculella* (Zeller) (Lepidoptera: *Gelechiidae*), is considered one of the most devastating insect pests on potatoes, feeding on roots, stems, foliage, and, more importantly, potato tubers. Severe infestations result in substantial yield and quality losses, even up to 100% losses to potato plants in fields as well as in storage [3]. *P. operculella* originated from the tropical mountainous region of South America and then distributed over a wide range of tropical and subtropical countries in North America, Europe, Africa, and Asia [4,5]. It was first introduced to China in 1937 and found on tobacco in Yulin, Guangxi province [6]. To our knowledge, it is widely distributed in more than twenty provinces in China, including Yunnan, Guizhou, and Sichuan [7,8].

The tomato leafminer, *Phthorimaea absoluta* (Meyrick) (Lepidoptera: *Gelechiidae*), is also known as *Tuta absoluta*, another important invasive pest within the *Gelechiidae* family, causing destructive harm to Solanaceous crops including potatoes. The larvae primarily feed on the mesophyll of all aerial parts of the potato plant, leading to leaf damage and plant withering, and resulting in extreme yield losses and economic damage [9]. It originated from Peru and quickly spread across Europe, Africa, and Asia in the past decades [10,11,12,13], and is now found in more than 110 countries and regions [14]. Since its first detection in Xinjiang province in 2017 [15], *P. absoluta* has subsequently spread throughout China and has been detected in more than ten provinces, especially in Yunnan province [16,17].

*P.operculella* and *P. absoluta* are two closely related species that feed on potato plants, and it is difficult to distinguish them by morphological characteristics. Their larvae feed on the tubers and/or foliage of potato crops, which causes cumulative damage. Both pests can occur on potato crops at the same time, rendering control methods to keep them under economic damage thresholds important. The cultivation of dominant potato varieties [18], the use of sex pheromones [19], and deeper seed planting are recommended to prevent pest infestation. However, the application of chemical insecticides has been the mainstay of controlling *P. operculella* and *P. absoluta* in diverse agricultural production systems [8,14]. To date, only cypermethrin and lufenuron have been registered to control *P. operculella* in China [20], with no options available to control *P. absoluta*. Several applied problems, such as poor chemical selection and substandard application practices, have exacerbated control failures and contribute to the evolution of a variety of resistance mechanisms in response to insecticide selection. Frequent applications of insecticides of the same chemical class or mode of action potentially select for insecticide resistance [21].

Current knowledge supports that insecticide resistance is primarily driven by metabolic enzymes (i.e., cytochrome P450-monooxygenases (P450s)) and target-site mutations affecting insecticide binding such as mutations in Ace1 (acetylcholinesterase), VGSC (voltage-gated sodium channels), and RDL (GABA-gated chloride channel) [22,23,24], conferring resistance to organophosphates, carbamates, pyrethroids, and fiproles, respectively. Metabolic and target-site resistance may be observed simultaneously or isolated in a population under continuous selection pressure. Insecticide resistance mediated using P450s is highly prevalent and has been extensively documented in association with metabolic detoxification under insecticide selection [24,25,26]. Simultaneously, the studies on target-site mutations conferring insecticide resistance have seen increasing importance in recent decades [27,28,29,30,31,32,33,34] and are considered of major concern. It has been reported that mutations in Ace1 (G247S [28], A302S [29], F402V [30], F439W/Y [31]) lead to resistance against organophosphates (OPs) and carbamates (CBs). Mutations in VGSC (V410L [32], M918I + L1014F [33]) have been shown to confer resistance to pyrethroids (PYs). Additionally, mutations in RDL (A301S [34]) have been linked to resistance against dieldrin and fipronil.

Previous studies revealed the existence of insecticide resistance to organophosphates, carbamates, pyrethroids, and diamides in *P. operculella* and *P. absoluta* field populations [6,11,35,36,37,38,39,40,41]. However, there are much fewer reports on the molecular basis of insecticide resistance in *P. operculella* than in *P. absoluta.* As these two closely related species both cause damage to potato crops, this study detected their multiple genetic mutations related to insecticide resistance by PCR and investigated the distribution and frequencies of mutations in the southwest and northwest regions of China. We attempted to explore the resistance mechanism in field-collected populations of *P. operculella* and *P. absoluta* with a specific focus on target-site mutations conferring resistance to commonly used chemical classes of insecticides, including OPs, CBs, PYs, and fiproles. These data provide a basis for developing resistance management strategies in potatoes, covering both pests under applied conditions.

## 2. Methods

### 2.1. Insects

*P. operculella* and *P. absoluta* larvae were collected at six different locations in China during April and August 2023 (Appendix A). Field-collected insects were stored in 100% ethanol, brought to the laboratory, and kept at −20 °C until use. The species determination was based on PCR and sequencing according to Kwon et al. [42].

### 2.2. Genomic DNA Extraction

Thirty individuals from each sampling site were selected randomly. The genomic DNA (gDNA) of individual larvae was prepared according to the method of Rinkevich et al. [43]. Briefly, the larva was placed in a tube with 500 µL of lysis buffer containing 100 mM Tris-HCl (pH = 8.0), 50 mM NaCl, 10 mM EDTA, 1% (*w*/*v*) SDS, and 0.1 mg/mL proteinase K, homogenized using a plastic pestle, and then incubated at 56 °C for 1 h. The mixture was set in an ice bath for 10 min after the addition of 50 µL 8 M potassium acetate, and then centrifuged at 12,000× *g* for 10 min. The supernatant was then transferred to a new tube and the gDNA was precipitated by adding ice-cold ethanol (2:1) at room temperature for 10 min. The pellet was washed with 75% ethanol and resuspended in ddH_2_O. The prepared gDNA samples were stored at −20 °C.

### 2.3. PCR Amplification of Target Gene Fragments and Sanger Sequencing

Fragments of the *PoVGSC* (~2500 bp), *PoAce1* (~3500 bp), *PoRdl* (~650 bp), *PaVGSC* (~1600 bp), *PaAce1* (~3500 bp), and *PaRdl* (~1000 bp) genes were amplified by PCR using primer pairs shown in Appendix A. The PCR reactions were performed with 2× Phanta Max Master Mix (Vazyme Biotech Co., Ltd., Nanjing, China) in a T-100 thermal cycler (Bio-Rad, Hercules, CA, USA) as follows: 35 cycles of 5 s at 98 °C, 10 s at 55 °C, and 30 s at 72 °C. The purified PCR products from heterozygotes were ligated with the pUC19 vector and transformed into competent cells of the *Escherichia coli* DH5α strain (Shenzhen Kangti Life Technology Co., Ltd., Shenzhen, China). The PCR products from homozygotes and three clones from heterozygotes were Sanger-sequenced by Sangon Biotech (Guangzhou, China).

### 2.4. DNA Sequence Analysis and Phylogenetic Analysis

DNA sequencing data were checked and curated manually. All confirmed sequences from filed populations, combined with data from the NCBI database, were aligned with MUSCLE [44]. The haplotypes were constructed using DnaSPv5 software from the alignment data [45]. The genealogical relations among haplotypes were analyzed using Networkv10 [46]. The phylogenetic tree was generated using MEGA X [47] with the maximum likelihood method and the Tamura 3-parameter model with 1000 bootstrap replicates based on the DNA alignment.

### 2.5. Molecular Clock Estimates

In order to examine the relationships among *Gelechiidae*, the mitochondrial cytochrome c oxidase subunit I (COI) gene sequences were sourced from the GenBank database (Appendix A). The divergence time estimation was performed using the RelTime method following a standard protocol as implemented in MEGA X under the GTR + G model with 1000 bootstrap replicates [47]. *Drosophila melanogaster* was set as an outgroup. There were a total of 1531 positions in the final dataset. The calibration was set to 101.9–125.2 MYA (*Phthorimaea operculella* versus *Helicoverpa armigera*), 79.8–127.3 MYA (*Phthorimaea operculella* versus *Plutella xylostella*), and 38–60.3 MYA (*Helicoverpa armigera* versus *Spodoptera frugiperda*) following the TimeTree database (http://timetree.org/, accessed on 12 February 2024). The other parameters were defined by default.

### 2.6. Syntenic Relationship of the Insecticide Target Gene Loci

To detect the collinearity of target genes between *P. operculella* and *P. absoluta*, the local comparative genomics analysis was conducted with TBtool software using the Genome Region Compare Suite plugin with the following blastp parameters: maxHsp500 and minLen100 [48]. The complete genome of *P. operculella* (GenBank assembly number: GCA_024500475.1) and *P. absoluta* (GenBank assembly number: GCA_027580185.1), along with their annotation information, were downloaded from the NCBI dataset (https://www.ncbi.nlm.nih.gov/genome/, accessed on 2 February 2024).

## 3. Results

### 3.1. Distribution and Frequency of Insecticide Resistance-Conferring Mutations

Three target genes expressing commonly known insecticide targets were analyzed for mutations conferring insecticide resistance. Based on previous studies, the mutations G119S, A201S, G328A, and F331W (*Torpedo californica* numbering) in *Ace1* were particularly analyzed in amplified sequences containing Ace1 codons 119–331 of *P. operculella* (*Po*Ace1) and *P. absoluta* (*Pa*Ace1). An amplified fragment of around 700 bp was analyzed from each species and, in total, four mutation sites were detected, i.e., F158Y, A201S, L231V, and F290V, including two formerly validated Ace1 target-site mutations (A201S and F290V). The frequencies of the detected mutant alleles are detailed in Table 1. Interestingly, the F158Y allele was exclusively detected in Ace1 of *P. operculella*, whereas the two known resistance mutations (A201S and F290V) were predominantly detected in *P. absoluta*. The commonly known Ace1 mutations G119S and F331W were not detected in field populations of either *P. operculella* or *P. absoluta*.

The *VGSC* gene of *P. operculella* and *P. absoluta* contains 31 and 34 exons, respectively, and encodes the target for pyrethroid insecticides. Here, we analyzed an amplified region (~2000 bp) containing codons 918 and 1014 of VGSC for the presence of mutations formerly described to confer pyrethroid resistance. Four common mutations (M918T, L925M, T929I, and L1014F, *Musca domestica* numbering) were observed in *P. absoluta* field populations at variable frequencies (Table 2). Among these, the M918T and T929I resistance alleles were shown to be widely distributed in sampled larvae of *P. absoluta* but completely absent in field populations of *P. operculella* (Table 2). The only known VGSC mutation detected in *P. operculella* was L1014F at quite low frequency, whereas the frequency of L1014F in *P. absoluta* was 100%.

Lastly, an approximately 750 bp fragment of the *RDL* sequence encompassing the A301S mutation (*D. melanogaster* numbering) was examined. This mutation was not found in *P. operculella*, but in *P. absoluta*. The frequency of the resistance allele A301S was 27%, 25.5%, and 34.5% in larvae collected at GD, PL, and XS sites, respectively (Table 3).

### 3.2. Haplotype Identification and Phylogeny

Based on the sequence data from field populations (30 individuals for each population) and the NCBI database, 6 *PoAce1* (Figure 1) and 4 *PaAce1* (Figure 2) haplotypes were identified. Fifteen synonymous and two non-synonymous mutations were identified in *P. operculella* (Figure 1A). The sequence of *PoAce1-*H1 originated from a laboratory population without exposure to any insecticides for decades. Blast searches revealed that haplotype *PoAce1*-H2 shared an identical sequence with the sequence (access number: MG265690.1) from Syria and all three field populations (Figure 1B,C). The *PoAce1*-H2 and *PoAce1*-H5 were found with no mutation in combination with either F158Y or A201S (Figure 1B,C). Similarly, two synonymous and three non-synonymous mutations were observed in *P. absoluta* (Figure 2A). The *PaAce1*-H1 was a common haplotype in China, the USA (access number: JARADA010000055.1, JAQAHW010000022.1, SNMR01040031.1), and Brazil (access number: KU985167.1) (Figure 2B). It was assumed to be an ancestral haplotype based on network results (Figure 2C).

For a better understanding, the haplotype identification and evolutionary origin of codons 918 and 1014 in VGSC were analyzed separately. A fragment of 569 bp containing VGSC codon 918 from 90 individuals in *P. operculella* was grouped into one haplotype. Nine haplotypes were identified with a shorter length (55 bp) of *PaVGSC.* Three non-synonymous mutations (M918T, L925M, and T929I) and one synonymous mutation in the encoding region were found (Figure 3A). The haplotypes PoVGSC-H1, PoVGSC-H2, and PoVGSC-H4 were observed in all field populations, also partially shared with the USA (access number: JARADA010000037.1, JAQAHW010000009.1, SNMR01042398.1) and Iran (access number: KY767010.1) haplotype (Figure 3B). Based on the phylogenetic tree, the M918T mutation has a different evolutionary origin (Figure 3B). As to codon 1014, a fragment of 340 bp containing VGSC codon 1014 from 90 individuals in *P. operculella* was grouped into three haplotypes. Three SNPs (single-nucleotide polymorphisms) were observed in the intron region, and only one non-synonymous mutation (L1014F) was identified. Similarly, the length of 414 bp containing the VGSC codon 1014 from 90 individuals in *P. absoluta* was grouped into one haplotype.

Only one haplotype of the *RDL* gene was identified from 90 individuals in *P. operculella*, with a length of 567 bp. In *P. absoluta,* one non-synonymous mutation in the encoding region and two polymorphic sites in the intron were observed (Figure 4A). From a total of 126 sequences obtained from 90 individuals, 5 haplotypes were identified. Of them, *PaRDL*-H2 was detected in all populations, including the USA (ARADA010000003.1, JAQAHW010000001.1, SNMR01042068.1), while *PaRDL*-H4 was uniquely distributed in GD city (Figure 4B,C). The phylogenetic tree and network results indicated that haplotypes with the A301S mutation were clustered into different branches (Figure 4B,C), suggesting their multiple evolutionary origins.

### 3.3. The Divergence Time among Gelechiidae

Based on the phylogenetic timetree, the divergence time of *Gelechiidae* was estimated due to the absence of fossil records. It has illuminated the temporal aspects of the evolutionary history of *P. operculella* and *P. absoluta*, around 46.56 million years ago (MYA) (Figure 5). Remarkably, *P. absoluta* exhibited an early divergence of approximately 41.27 MYA from *P. operculella* (36.51 MYA).

### 3.4. The Collinearity of Specific Genomic Regions in P. operculella and P. absoluta

Three target genes, *VGSC* (*Po*Scaffold2 versus *Pa*Scaffold9), *RDL* (*Po*Scaffold3 versus *Pa*Scaffold1), and *Ace1* (*Po*Scaffold113 versus *Pa*Scaffold22), were located in different genomic regions. It appeared to be the orthologous genomic regions that span 280~410 kb between *P. operculella* and *P. absoluta* with high collinearity. Interestingly, some gene duplications, insertions or deletions, and rearrangements were observed between *P. operculella* and *P. absoluta* (Figure 6).

## 4. Discussion

Research on insecticide resistance has proliferated since the 1960s [49], and investigators worldwide are striving to unravel the mechanisms driving the evolution of insecticide resistance. The most rapid and convenient way to detect the resistance levels in field populations is to genotype field-collected samples for known target-site mutations against commonly used classes of chemistry, thus allowing for the mapping of the distribution and allele frequencies of resistant haplotypes. *P. operculella* and *P. absoluta* are two major moth pests that feed on potato plants and particularly *P. absoluta* has been shown to evolve target-site resistance against several classes of insecticides [37,40,41], whereas such knowledge for *P. operculella* is largely lacking. Phylogenetic reconstructions suggested that *P. absoluta* and *P. operculella* diverged approximately 46.56 MYA (Figure 5). The orthologous genomic regions, including the target gene, between *P. operculella* and *P. absoluta* showed high collinearity (Figure 6). However, the target-site mutations exhibited unexpected genetic divergence under insecticide selection pressure.

Mutations in the AChE leading to insensitivity towards OP and CB insecticides are known and largely contribute to OP and CB field resistance in numerous pest insects [50]. Insensitive AChE is conferred by one or more point mutations in the *Ace1* gene. Such mutations include, for example, G119S [28], A201S [51], A280T [31], G328A [31], and F331C/Y/W [31,52], to name those most reported. Several of these mutations have been functionally validated in recombinantly expressed wildtype and mutant AChE variants [53]. These results showed a strong correlation between frequencies of the mutation and phenotypic levels of resistance. Among them, A201S has been reported in field-collected *P. absoluta* populations from Brazil and Europe [41] and is known to confer OP and CB resistance. In our survey, the A201S mutation was also observed in field-collected Chinese *P. absoluta* samples at high frequency without susceptible homozygous haplotypes (Table 1). Although we did not have bioassay results, the extremely high conservation of the alanine (A201) in all other insect species and previous reports of the same mutation in other OP-resistant insects strongly suggest that this mutation also confers resistance in *P. absoluta*. Additionally, the F290V mutation was also found in Yunnan and Xinjiang provinces. Several studies functionally validated that these mutations confer insecticide resistance, thus suggesting that the application of OPs and CBs is likely to be ineffective in controlling Chinese populations of *P. absoluta.* Compared with *P. absoluta*, only A201S was observed in *P. operculella* field populations, and even absent in the XW colony, among these common mutations about Ace1 related to insecticide resistance. Du et al. found that *P. operculella* collected from Yunnan Province developed significant resistance to dichlorvos, methamidophos, and ethyl-parathion in 2006 [6]. In view of the correlation between mutations in *Ace1* and OP resistance, the A201S mutation may be one of the reasons for mediating OP resistance in *P. operculella* in Yunnan Province. Intriguingly, the F158Y mutation was only observed in *P. operculella*, whereas there is no previous report of the F158Y mutation. Whether the natural F158Y mutation is related to insecticide resistance remains unclear and needs to be clarified in the future.

Pyrethroids, a major class of neurotoxic insecticides, have been extensively used to control various agricultural pests. Among these, cypermethrin is most widely used to control *P. operculella* in China [20]. Resistance to PYs often arises through knockdown resistance (kdr) resulting from point mutations in the VGSC that reduce the sensitivity of the insect nervous system to these compounds [54]. Three target resistance-conferring VGSC mutations, including kdr (L1014F), kdr-his (L1014H), and super-kdr (M918T + L1014F), have been identified in multiple species, such as *Diptera* [55], *Lepidoptera* [56], and *Siphonaptera* [57]. Based on our survey, the kdr and super-kdr mutations have been commonly detected in *P. absoluta* field populations. The construction of the haplotype network and phylogeny analysis indicated that the substitution of methionine (M918) to threonine (T918) has different origins (Figure 3). These indications suggest that the *P. absoluta* in China would exhibit resistance to PYs. Additionally, the T929I mutation was observed with relatively high frequencies in *P. absoluta* field populations. The T929I mutation was first reported in the pyrethroid-resistant diamondback moth, *P. xylostella*, and is presumed to play an important role in pyrethroid resistance [58]. T929I in combination with L1014F was found to make the sodium channel highly insensitive to a range of type I and type II pyrethroid insecticides and DDT [59]. The combination of M918T, T929I, and L1014F mutations was observed in *P. absoluta* field populations from Yunnan and Xinjiang provinces and reported in populations from Europe and Brazil [40]. Generally, the combined mutations often result in higher resistance levels, affecting a broad spectrum of pyrethroids. Our findings suggested that *P. absoluta* has historically undergone strong selection with a range of pyrethroid insecticides, resulting in significant control problems under applied conditions in the field. In *P. operculella*, the kdr mutation was at low frequency in field populations, with other mutations associated with PY resistance barely present. Long-term use of chemical insecticides over decades has led to ineffective pest control, resulting in widespread distribution throughout China. It is inferred that P450s may play an important role in insecticide resistance in *P. operculella* field populations in China.

GABA receptors, targeted by cyclodienes and fiproles, are ligand-gated homopentameric chloride channels consisting of RDL subunits encoded by the *RDL* gene. One common target-site mutation is A301S, conferring high levels of cyclodiene resistance and lower levels of fiprole resistance, e.g., in *Nilaparvata lugens* [34]. Here, we detected the A301S mutation at moderate frequency in *P. absoluta* field populations, but not in *P. operculella* field populations. The A301S mutation in three *P. absoluta* field populations was narrowly distributed at a low frequency, and its contribution to insecticide failures in tomato leafminer control remains unclear.

Overall, we detected at varying frequencies multiple target-site mutations in *P. operculella* and *P. absoluta* field populations in China likely to compromise the efficacy of commonly used chemical classes of insecticides such as organophosphates and pyrethroids. Nevertheless, *P. operculella* showed a much lower frequency of target-site mutations than *P. absoluta*, suggesting that the latter is more difficult to control with less costly or less expensive insecticides such as pyrethroids. However, we have not investigated the potential contribution of upregulated metabolic detoxification mechanisms which can modulate and contribute substantially to target-site mediated resistance to pyrethroids [60]. The current control strategy in most areas sampled in China involves spraying insecticides to protect the potato crops, but some of the recommended chemical classes such as pyrethroids and OPs are likely to fail under field conditions, particularly against *P. absoluta*. Therefore, resistance monitoring, such as that shown here by genotyping samples for the presence of target-site mutations, is crucial for the implementation of rational strategies of insecticide application and resistance management.

## Figures and Tables

**Figure 1 insects-15-00194-f001:**
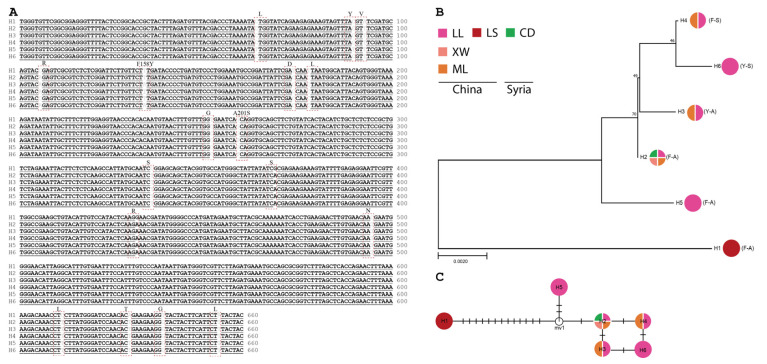
The evolution of the *Ace1* haplotypes in different populations of *P. operculella.* (**A**) The nucleotide alignment of six haplotypes of *Ace1*, H1–H6, is presented. Mutations are labeled in red dotted boxes. (**B**) The phylogenetic tree is inferred using the maximum likelihood method. Bootstrap values are conducted for 1000 replicates and only scores above 50 are shown. The abbreviations of locations are LL (Luliang), XW (Xuanwei), ML (Malong), LS (laboratory population) from China, and CD (Damascus) from Syria. (**C**) The network presents the genealogy of *Ace1* haplotypes, and the short solid lines indicate the possible mutational steps between haplotypes.

**Figure 2 insects-15-00194-f002:**
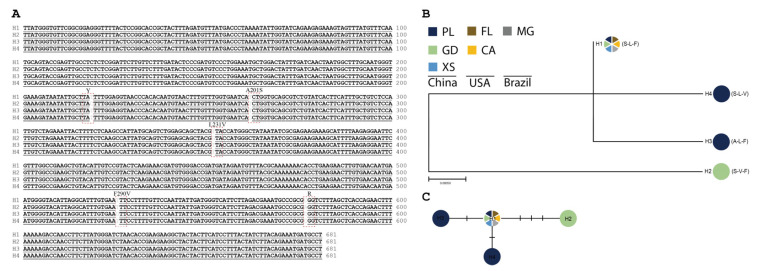
The evolution of the *Ace1* haplotypes in different populations of *P. absoluta.* (**A**) The nucleotide alignment of four haplotypes of *Ace1*, H1–H4, is presented. Mutations are labeled in red dotted boxes. (**B**) The phylogenetic tree is inferred using the maximum likelihood method and presented. Bootstrap values are conducted for 1000 replicates and only scores above 50 are shown. The abbreviations of locations are PL (Panlong), GD (Guandu), and XS (Xinshi) from China, FL (Florida) and CA (California) from the USA, and MG (Minas Gerais) from Brazil. (**C**) The network presents the genealogy of *Ace1* haplotypes, and the short solid lines indicate the possible mutational steps between haplotypes.

**Figure 3 insects-15-00194-f003:**
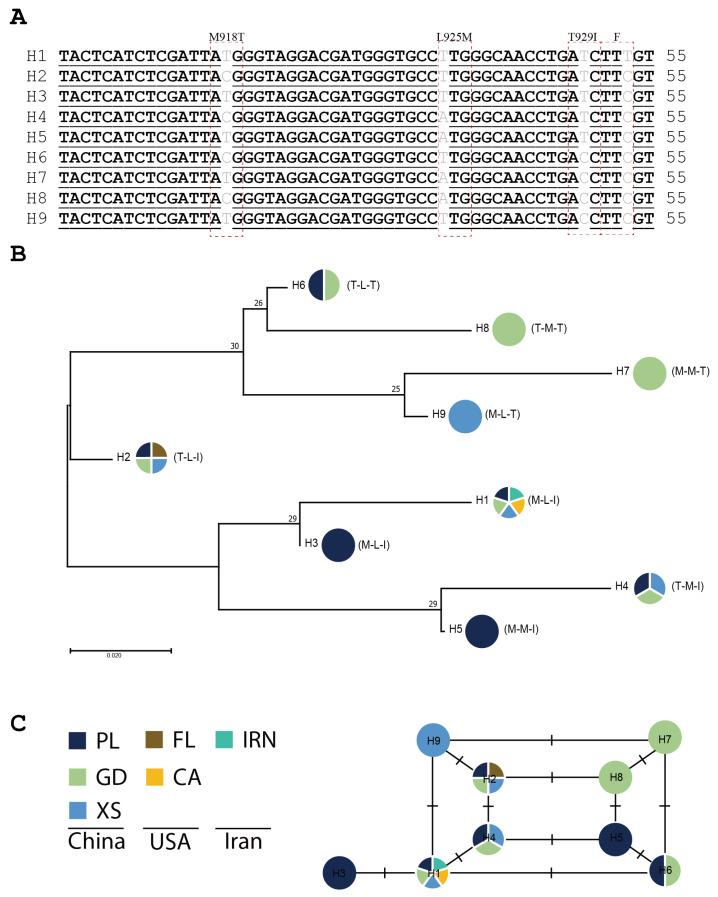
The evolution of the *VGSC* haplotypes in different populations of *P. absoluta.* (**A**) The nucleotide alignment of nine haplotypes of *VGSC* containing codon 918, H1–H9, is presented. Mutations are labeled in red dotted boxes. (**B**) The phylogenetic tree is inferred using the maximum likelihood method. Bootstrap values are conducted for 1000 replicates and only scores above 50 are shown. The abbreviations of locations are PL (Panlong), GD (Guandu), and XS (Xinshi) from China, FL (Florida) and CA (California) from the USA, and IRN (Iran) from Iran. (**C**) The network presents the genealogy of *VGSC* haplotypes, and the short solid lines indicate the possible mutational steps between haplotypes.

**Figure 4 insects-15-00194-f004:**
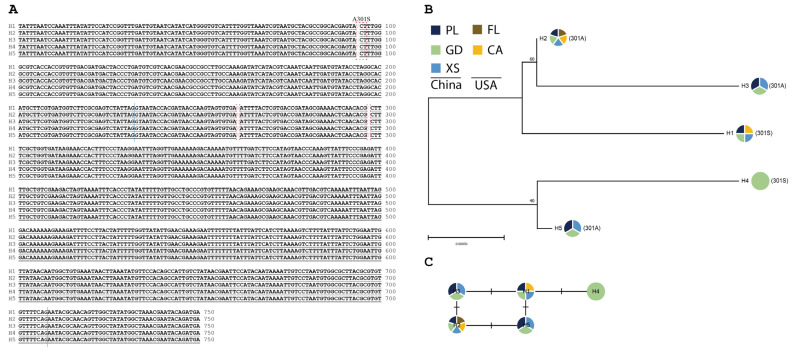
The evolution of the *RDL* haplotypes in different populations of *P. absoluta.* (**A**) The nucleotide alignment of five haplotypes of *RDL*, H1–H5, is presented. Mutations are labeled in red dotted boxes. The intron region is marked in a blue vertical line, and the GT-AG is presented in italics. (**B**) The phylogenetic tree is inferred using the maximum likelihood method. Bootstrap values are conducted for 1000 replicates and only scores above 50 are shown. The abbreviations of locations are PL (Panlong), GD (Guandu), and XS (Xinshi) from China, and FL (Florida) and CA (California) from the USA. (**C**) The network presents the genealogy of *RDL* haplotypes, and the short solid lines indicate the possible mutational steps between haplotypes.

**Figure 5 insects-15-00194-f005:**
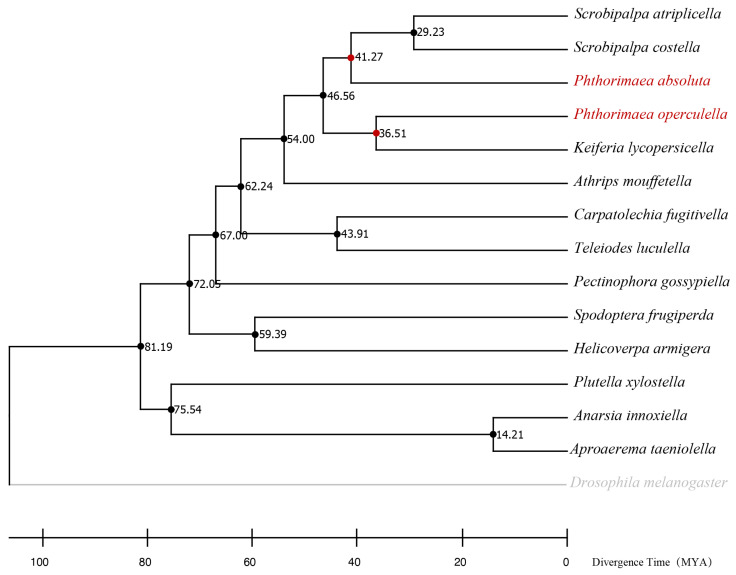
Timetree analysis for Gelechiidae members using the RelTime method. *Drosophila melanogaster* was used as an outgroup. *Helicoverpa armigera*, *Spodoptera frugiperda,* and *Plutella xylostella* were set as calibration for Gelechiidae members lacking fossil histories. The divergence time of *P. operculella* and *P. absoluta* was marked in red circle. The red font represents the two target species of this study, The gray font represents the use of Drosophila mela-nogaster as an outgroup.

**Figure 6 insects-15-00194-f006:**
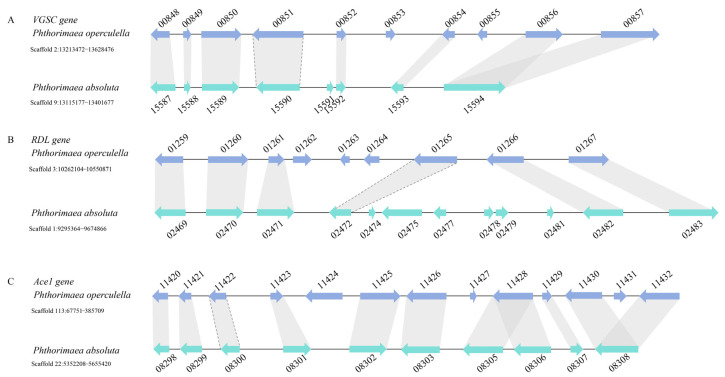
The collinearity map depicting synteny of target gene loci in *P. operculella* and *P. absoluta.* The complete genomes of *P. operculella* and *P. absoluta* are available under the GenBank assembly numbers GCA_024500475.1 and GCA_027580185.1, respectively. The genomic region, 280~410 kb, analyzed included *Ace1*, *VGSC*, and *RDL*. The obtained synteny between *P. operculella* (marked in marine blue) and *P. absoluta* (marked in tiffany blue) is shown in (**A**), (**B**), and (**C**), respectively. The arrow indicates the direction of protein coding. The synteny of target genes (*Ace1*, *VGSC*, and *RDL*) is marked in a dashed line.

**Table 1 insects-15-00194-t001:** Frequencies of *Ace1* genotypes in different field populations of *P. operculella* and *P. absoluta*.

Species	Populations	* n *	F158Y			A201S			L231V			F290V		
			F	F/Y	Y	A	A/S	S	L	L/V	V	F	F/V	V
*P. operculella*	LL	30	83	10	7	87	13	0	0	0	100	100	0	0
	ML	30	86	7	7	90	7	3	0	0	100	100	0	0
	XW	30	90	10	0	100	0	0	0	0	100	100	0	0
*P. absoluta*	PL	30	100	0	0	0	77	23	77	23	0	40	60	0
	GD	30	100	0	0	0	67	33	63	37	0	73	27	0
	XS	30	100	0	0	0	63	37	97	3	0	70	30	0

Note: The abbreviations of locations are LL (Luliang), ML (Malong), XW (Xuanwei), PL (Panlong), GD (Guandu) and XS (Xinshi) of China. The abbreviation of amino acids are F (phenylalanine), Y (tyrosine), A (alanine), S (serine), L (leucine) and V (valine).

**Table 2 insects-15-00194-t002:** Frequencies of *VGSC* genotypes in different field populations of *P. operculella* and *P. absoluta*.

Species	Populations	* n *	M918T			L925M			T929I			L1014F		
			M	M/T	T	L	L/M	M	T	T/I	I	L	L/F	F
*P. operculella*	LL	30	100	0	0	100	0	0	100	0	0	84	13	3
	ML	30	100	0	0	100	0	0	100	0	0	87	10	3
	XW	30	100	0	0	100	0	0	100	0	0	77	20	3
*P. absoluta*	PL	30	30	47	23	93	7	0	23	47	30	0	0	100
	GD	30	53	40	7	90	10	0	10	47	43	0	0	100
	XS	30	50	37	13	93	7	0	13	40	47	0	0	100

Note: The abbreviations of locations are LL (Luliang), ML (Malong), XW (Xuanwei), PL (Panlong), GD (Guandu) and XS (Xinshi) of China. The abbreviations of amino acids are M (methionine), T (threonine), L (leucine), F (phenylalanine) and I (isoleucine).

**Table 3 insects-15-00194-t003:** Frequencies of *RDL* genotypes in different field populations of *P. operculella* and *P. absoluta*.

Species	Populations	* n *	A301S		
			A/A	A/S	S/S
*P. operculella*	LL	30	100	0	0
	ML	30	100	0	0
	XW	30	100	0	0
*P. absoluta*	PL	30	63	20	17
	GD	30	66	17	17
	XS	30	54	23	23

Note: The abbreviations of locations are LL (Luliang), ML (Malong), XW (Xuanwei), PL (Panlong), GD (Guandu) and XS (Xinshi) of China. The abbreviations of amino acids are A (alanine) and S (serine).

## Data Availability

The data supporting the conclusions of this article are provided within the article. The original datasets analyzed in this study are available from the corresponding author upon request. The nucleotide sequences reported in this study have been deposited in the GenBank database listed in Appendix A.

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
