# Peer review of "Presence of Multiple Genetic Mutations Related to Insecticide Resistance in Chinese Field Samples of Two Phthorimaea Pest Species"

_insects, 2024, doi:10.3390/insects15030194_

Round 1
Reviewer 1 Report
Comments and Suggestions for Authors
Insecticide resistance in agricultural pests is a growing concern worldwide, impacting the efficacy of pest management strategies and threatening food security. The study by Zhu et al., titled "Presence of multiple genetic mutations related to insecticide resistance in Chinese field samples of two Phthorimaea pest species," makes a significant contribution to the field by exploring the genetic underpinnings of insecticide resistance in two key pests: Phthorimaea operculella (potato tuber moth) and Phthorimaea absoluta (tomato leafminer). These species are notorious for their impact on potato and tomato crops, respectively, causing substantial economic losses. With the intensification of agriculture and the consequent reliance on chemical control measures, understanding the mechanisms behind insecticide resistance is crucial for developing effective and sustainable pests.
This study focuses on genetic mutations associated with resistance to common insecticide classes, such as organophosphates, carbamates, and pyrethroids, in field populations of P. operculella and P. absoluta in China. While the identification of target-site mutations provides valuable insights into potential resistance mechanisms, the study has significant gaps, notably the lack of phenotypic resistance data.
The major limitation of this study is the absence of phenotypic resistance data that correlates the detected genetic mutations with actual resistance levels in the field populations of the pests. Without this data, it is challenging to determine the real impact of these mutations on insecticide efficacy. The presence of mutations is indicative but not conclusive of resistance, as the functional expression of these mutations and their interaction with various insecticides could vary significantly. Phenotypic assays would provide direct evidence of resistance and help in understanding the fitness costs associated with these mutations, which is crucial for resistance management strategies.
The article briefly mentions the use of pyrethroids, organophosphates, and carbamates for controlling P. operculella and P. absoluta in China but does not provide detailed information on the specific compounds used, their application rates, or the history of their use. This information is critical to contextualize the selection pressure exerted on these pest populations. The evolution of resistance is directly influenced by the insecticide regime applied, including the frequency of applications, the rotation of chemical classes, and the adoption of integrated pest management (IPM) strategies. A detailed history of insecticide use would enable a better understanding of the selection pressures that led to the observed genetic mutations.
While the study identifies several mutations in the target-site genes Ace1, VGSC, and RDL, it does not thoroughly explore the association of these mutations with resistance to specific insecticides. For instance, the mutations A201S, L231V, and F290V in Ace1 and M918T, L925M, T928I, and L1014F in VGSC are mentioned, but the study does not detail which insecticides these mutations confer resistance to, or the level of resistance provided. This gap significantly limits the utility of the study for developing resistance management strategies, as it is unclear how these mutations affect the efficacy of current insecticide treatments.
Provide a comprehensive history of insecticide use for P. operculella and P. absoluta control in China, including the types of insecticides used, application regimes, and changes in use patterns over time.
Explore the functional significance of the identified mutations in relation to specific insecticides, potentially through bioassays or molecular studies, to directly link mutations with resistance phenotypes.
Recommendations to address these gaps, future studies should aim to:
Conduct phenotypic assays to link genetic mutations with actual resistance levels and understand the ecological fitness of resistant vs. susceptible populations. If you do not have this data, at least add detailed information on the actual level of resistance in each species to different insecticides in China. Integrating genetic data with phenotypic resistance levels and detailed insecticide use history would provide a more holistic understanding of resistance mechanisms and facilitate the development of more effective and sustainable pest management strategies.
Comments on the Quality of English Language
The manuscript needs thorough proofreading to correct typographical errors, misspellings, and minor grammatical mistakes.
Author Response
1.Conduct phenotypic assays to link genetic mutations with actual resistance levels and understand the ecological fitness of resistant vs. susceptible populations. If you do not have this data, at least add detailed information on the actual level of resistance in each species to different insecticides in China. Integrating genetic data with phenotypic resistance levels and detailed insecticide use history would provide a more holistic understanding of resistance mechanisms and facilitate the development of more effective and sustainable pest management strategies.
Response:
Thank you for your excellent comments and suggestions, which are greatly appreciated. As suggested, we have added the registered insecticides to control two pests, as detailed in lines 90-92, and revised the discussion. Long-term use of chemical insecticides over decades has led to ineffective pest control, resulting in widespread distribution throughout China. This study aims to rapidly determine the level of insecticide resistance in field populations. We attempted to investigate the mechanism of resistance in field-collected populations of P. operculella and P. absoluta, with a specific focus on target site mutations conferring resistance to commonly used chemical classes of insecticides. Previous numerous reports have been largely confirmed and linked to insecticide resistance in other species [27-33]. Their results showed a strong correlation between the frequency of the mutation and the phenotypic level of resistance. The same mutation in other insecticide-resistant insects strongly suggests that the target gene mutations in our survey also confer resistance. The phenotypic resistance data are undoubtedly the best evidence of insecticide resistance. However, the samples collected for this survey were preserved in 100% ethanol until used. It’s difficult to conduct bioassay experiments.
2.The manuscript needs thorough proofreading to correct typographical errors, misspellings, and minor grammatical mistakes.
Response:
Thank you for your suggestion. We have thoroughly conducted a comprehensive review of the entire manuscript to ensure all spelling and other errors have been corrected.
The revised manuscript is uploaded. Please refer to the attachment for details.
Reviewer 2 Report
Comments and Suggestions for Authors
The authors studied the three common insecticide targets in P. operculella and P. absoluta. They used different samples to clone and sequence gene Ace1, VGSC and RDL. Their results could have potential usage in pest control in potatoes.
comments:
1. How many copies of Ace1, VGSC and RDL in P. operculella and P. absoluta?
2. It is better to revise the discussion. For example, the author can explain the result 3.1 (from line 179 to line 183).
Comments on the Quality of English LanguageThe quality of English Language is OK.
Author Response
- How many copies of Ace1, VGSC and RDL in P. operculella and P. absoluta?
Response:
The blast search using Ace1, VGSC and RDL gene sequences from Helicoverpa armigera suggests that all three target genes in P. operculella and P. absoluta are single-copy (reference genomes: GCA_024500475.1 for P. operculella and GCA_027580185.1 for P. absoluta), which is also supported by the collinearity result of specific genomic regions in P. operculella and P. absoluta.
- It is better to revise the discussion. For example, the author can explain the result 3.1 (from line 179 to line 183).
Response:
Thank you for your suggestion. The discussion section of the manuscript has been revised thoroughly, and the explanation of the result 3.1 were added to line 521-530.
The revised manuscript is uploaded. Please refer to the attachment for details.

Round 2
Reviewer 1 Report
Comments and Suggestions for Authors
Thank you for submitting the reviewed version of the manuscript. The authors have satisfactorily addressed the corrections and suggestions. Just a typo in line 465.
Comments on the Quality of English LanguageA typo in line 465